# CARD: CertifiAble Reweighting for Single Domain Generalization Object Detection

## Abstract

Single Domain Generalization Object Detection (S-DGOD) is a challenging yet practical task, where we only have access to data from one specific source domain to train an object detection network, but have to generalize to numerous unseen target domains. Recent works point out that the learning dynamics of Deep Neural Networks (DNNs) are biased by gradient descent to learn simple semantics, which are usually non-causal and spuriously correlated to the ground truth labels, as a result, DNN-based object detection networks fail to consistently generalize well in the Out-of-Domain (OoD) scenario. In this paper, we focus on S-DGOD based on theoretical analysis, exploring a classic and widely-used approach, Generalizable Reweighting (GRW), which iteratively reweightes the training samples to improve generalization performance. In our theoretical analysis, we first identify that the vanilla GRW hardly outperforms Empirical Risk Minimization (ERM) in the S-DGOD scenario. To provide a generalization guarantee, we further derive Certifiable Feature Perturbation (CFP) based on our theory, which aims to train a robust object detection network against additional perturbations added to the extracted features. We demonstrate that GRW works well with CFP in achieving OoD generalization, thus, surpassing ERM by a large margin under worse conditions. This brand new reweighting strategy is named Certifiable Reweighting (CARD). Our extensive experiments show that the proposed CARD achieves SOTA performance compared to baseline methods on the five urban-scene S-DGOD benchmarks.

## 1 Introduction

Object detection (Ren et al., 2015; Redmon et al., 2016; Lin et al., 2017a; Tan & Le, 2019; Zhu et al., 2020; Dai et al., 2021) is a fundamental task in computer vision. However, previous works lack thorough theoretical and systematic analysis on their generalization studies, especially the cases where networks to generalize to unseen test data drawn from the test distribution distinct from the training distribution. Recently, researchers (Pan et al., 2018; 2019; Huang et al., 2019; Wu & Deng, 2022; Vidit et al., 2023; Rao et al., 2023) have been dealing with a realistic task, Single Domain Generalization Object Detection (S-DGOD), which aims to train an object detection network on single source domain data and then generalize to multiple unseen target domains, to improve object detectors' generalization ability. Pezeshki et al. (2021); Krueger et al. (2021); Huang et al. (2022) point out that the failure of Deep Neural Networks (DNNs) in generalization is due to the learning bias of DNNs under gradient descent, where DNNs exhibit an inclination to make predictions based on the easy-to-learn features, such as backgrounds.

A classical yet effective technique to cope with this is Generalizable Reweighting (GRW), which aims to reweight the training samples to rebuild a weighted loss (Shimodaira, 2000; Fang et al., 2020; Sagawa et al., 2019; Krueger et al., 2021), forcing DNNs to upweight the influence of those samples with predictive features during training DNNs. However, we identify and theoretically demonstrate that the DNN trained by vanilla GRW is hard to overwhelm the one trained by Empirical Risk Minimization (ERM). Theory can be found in Section 4. In our theory, when a network is over-parameterized, as training risk $\hat{\mathcal{R}} \to 0$, both the networks trained by ERM and GRW converge to the same networks. The intuition is that when the training samples carry sufficient features, i.e., all the features in the hidden feature space can be represented by the interpolation between the given training features, then there is exactly one optimal network. Thus, when $\hat{\mathcal{R}} \to 0$, both

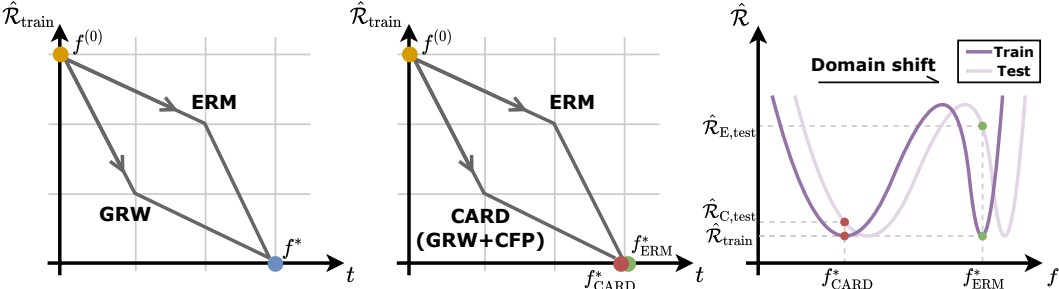

Figure 1: **Illustration of the failure of GRW and how CARD works.** *Left*: As time $t \to \infty$ and training risk $\hat{\mathcal{R}} \to 0$, both ERM and GRW converge to the same solution $f^*$, although the training dynamics are going along two different paths. Thus, GRW is hard to improve over ERM in the OoD scenario. *Middle*: With the utilization of CFP on GRW, ERM and GRW converge to different parameters, $f^*_{\text{ERM}}$ and $f^*_{\text{CARD}}$, respectively. *Right*: Our proposed CARD is demonstrated to be robust against perturbations, and have a flat minima on training risk compared to ERM. Thus, the network $f^*_{\text{CARD}}$ trained on the source domain generalizes better than $f^*_{\text{ERM}}$ and the corresponding risk on target domains $\hat{\mathcal{R}}_{\text{C,test}}$ is lower than $\hat{\mathcal{R}}_{\text{E,test}}$.

ERM and GRW converge to the same parameter although they are optimized along different paths. See Figure 1 (left) for illustration.

To address this, we propose Certifiable Feature Perturbation (CFP) to guide the learning dynamics of GRW, and the overall algorithm is named Certifiable Reweighting (CARD). In Section 5.1, we theoretically demonstrate that CFP helps the converged network trained by GRW prevail over the one trained by ERM. Our theory shows that with the help of CFP, as the training risk $\hat{\mathcal{R}} \to 0$, networks trained by ERM and GRW converge to different $f^*_{\text{ERM}}, f^*_{\text{CARD}}$, respectively, where $f^*_{\text{CARD}}$ depends not only on the training samples but also their weights. See Figure 1 (middle) for illustration. In this case, the learning dynamics of parameters are guided by the reweighted training samples, eliminating the learning bias of DNNs to improve generalization. Meanwhile, we theoretically demonstrate in Section 5.3 that the network $f^*_{\text{CARD}}$ is more robust against perturbations, which helps find the flat minima, thus, having lower test risk $\hat{\mathcal{R}}_{\text{test}}$ than $f^*_{\text{ERM}}$. See Figure 1 (right).

Our main contributions can be summarized as follows:

1. To the best of our knowledge, we are the first to identify and demonstrate that the vanilla GRW fails to overwhelm ERM in the S-DGOD task, which pertains to both classification and regression simultaneously.

2. To address this, we propose CARD, a methodology with theoretical guarantees that certifies the effectiveness of the converged network trained by GRW with CFP in achieving OoD generalization.

3. Extensive experiments demonstrate our algorithm stemmed from CARD empirically outperforms previous SOTA baselines on the challenging urban-scene S-DGOD benchmarks.

## 2 RELATED WORKS

### 2.1 GENERALIZABLE REWEIGHTING

To improve robustness and generalization performance, researchers have proposed various classic yet effective reweighting strategies. The most popular one is Importance Weighting (IW) (Shimodaira, 2000; Fang et al., 2020), which reweights training samples by quantifying their importance and leads to weighted empirical training loss. The rest of the works can be categorized into static (Shimodaira, 2000; Sagawa et al., 2020; Cui et al., 2019; Cao et al., 2019; Liu & Chawla, 2011) and dynamic methods (Wen et al., 2014; Zhai et al., 2021b; Michel et al., 2021; Zhai et al., 2021a; Lahoti et al., 2020; Michel et al., 2022; Lin et al., 2017b; Sagawa et al., 2019; Shu et al., 2019; Han et al., 2023; Krueger et al., 2021).

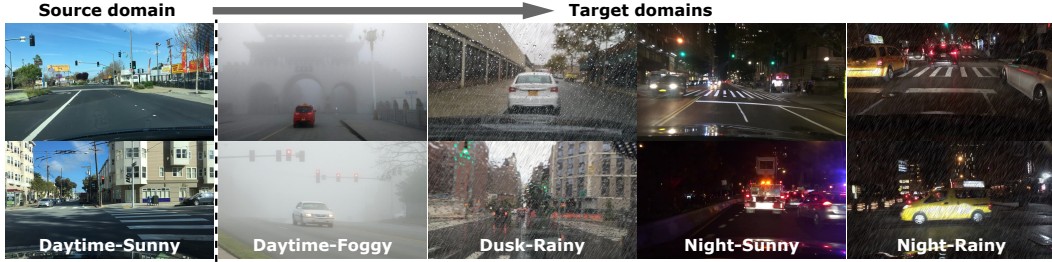

Figure 2: **Samples of the S-DGOD datasets.** The dataset contains 5 domains: Daytime-Sunny (source), Daytime-Foggy, Dusk-Rainy, Night-Sunny and Night-Rainy. S-DGOD aims at finding a network trained on a single source domain, which generalizes well to the multiple target domains.

**Static methods.** This kind of method primarily employs static weights which depend on the training distribution. Cui et al. (2019) propose to quantify the data overlap by a small neighboring region associated with each training sample to discover the effective number of samples for each class to reweight the training loss. Cao et al. (2019) proposes label-distribution-aware margin (LDAM) loss together with reweighting to deal with the category imbalance.

**Dynamic methods.** On the contrary, dynamic methods adaptively adjust the weights based on the learning dynamics for each training sample. Sagawa et al. (2019) aims to determine the worst-case and minimize their loss over the worst groups in the training data, which can be seen as a special case of IW where the weights of worst-cases are updated based on the current loss. Krueger et al. (2021) reweighs the training loss for each sample by adding a penalty on the variance of training risks. Han et al. (2023) proposes a straightforward framework employing IW on mixed-up samples, mitigating overfitting in over-parameterized models, and enhancing robustness against subpopulation shifts.

While the idea of GRW has been widely used, our study reveals that the vanilla GRW struggles to surpass ERM in S-DGOD.

## 2.2 SINGLE DOMAIN GENERALIZATION OBJECT DETECTION

S-DGOD aims at training object detectors on a single source domain and then generalizing to numerous target domains, see Figure 2 for details. Previous methods for S-DGOD can be generally categorized into feature normalization (Pan et al., 2018; Huang et al., 2019; Fan et al., 2021), and invariant-based algorithms (Choi et al., 2021; Wu & Deng, 2022; Zhao et al., 2022).

**Feature normalization.** IBN-Net (Pan et al., 2018) proposes to integrate Instance Normalization (IN) and Batch Normalization (BN) into DNNs for effective generalization. Iterative Normalization (Huang et al., 2019) leverages Newton's iterations to perform feature normalization in an iterative manner. ASRNorm (Fan et al., 2021) proposes adaptive standardization and rescaling normalization for improving single domain generalization performance.

**Invariant-based.** RobustNet (Choi et al., 2021) proposes an instance-specific whitening loss function to learn domain-specific features from domain-invariant feature representations. Cyclic-Disentangled Self-Distillation (Wu & Deng, 2022) tries to disentangle the domain-invariant representations using a distillation module. Style-Hallucinated Dual Consistency Learning (Zhao et al., 2022) introduces two key constraints, encouraging models to learn invariant representation across style-diversified samples.

**Others.** CLIPGap (Vidit et al., 2023) attempts to align the pre-trained knowledge of CLIP with the S-DGOD task. Their findings indicate that the size of the pre-training dataset significantly influences the enhancement of generalization capabilities.

While numerous approaches have been proposed for S-DGOD, we argue that the majority of these methods fundamentally adhere to the core idea of GRW. Most methods endeavor to facilitate the acquisition of more universally applicable features for predictions. In doing so, they implicitly change

the influence of different features in the learning process, predisposing DNNs towards acquiring generalizable attributes. Given that the experimental outcomes on S-DGOD benchmarks reveal these methods fall short of significantly surpassing ERM, this inspires us to investigate the underlying cause in this paper. We commence this inquiry by examining the vanilla GRW.

## 3 NOTATIONS

We denote each datum with its corresponding labels $(\mathbf{x}_i, \mathbf{y}_{1,i}, \mathbf{y}_{2,i})_{i=1}^n \in \{\boldsymbol{X}, \boldsymbol{Y}_1, \boldsymbol{Y}_2\}$ with categories labels $\mathbf{y}_{1,i}$ and bounding box labels $\mathbf{y}_{2,i}$. We use $P$ to denote the data distribution of $\{\boldsymbol{X}, \boldsymbol{Y}_1, \boldsymbol{Y}_2\}_P$ and $P_{\text{train}}, P_{\text{test}}$ to denote the training and testing distributions. Generally, object detection network $f(\cdot)$ contains a shared feature extraction network with parameters $\Theta$, a classification branch with parameters $\eta_1$, and a regression branch with parameters $\eta_2$. The classification and bounding box predictions are denoted as $\hat{\mathbf{y}}_1 = f(\mathbf{x}; \Theta, \eta_1)$ and $\hat{\mathbf{y}}_2 = f(\mathbf{x}; \Theta, \eta_2)$, respectively. Operation $\cdot$ denotes the matrix product. Empirical Risk Minimization (ERM) (Vapnik, 1999) trains a network via minimizing the empirical risk: $\hat{\mathcal{R}}_{P_{\text{train}}} = \frac{1}{n} \sum_{i=1}^n \hat{\mathcal{L}}_{\text{train}}(\mathbf{x}_i, \mathbf{y}_{1,i}, \mathbf{y}_{2,i}; \Theta, \eta_1, \eta_2)$, where training loss $\hat{\mathcal{L}}_{\text{train}}$ contains classification loss $\mathcal{L}_{\text{cls}}$ and regression loss $\mathcal{L}_{\text{reg}}$, $\hat{\mathcal{L}}_{\text{train}} = \mathcal{L}_{\text{cls}} + \mathcal{L}_{\text{reg}}$.

## 4 FAILURE OF GRW ON S-DGOD

As discussed in Section 2.2, we argue that various algorithms proposed for improving S-DGOD performance are indeed variants of RW. Instead of analyzing these algorithms individually, we consider a general form of what we call, Generalizable Reweighting (GRW):

$$\mathcal{L}_{\text{train}}(f) = \sum_i w_i \left( \mathcal{L}_{\text{cls}}(\mathbf{x}_i, \mathbf{y}_{1,i}; \Theta, \eta_1) + \mathcal{L}_{\text{reg}}(\mathbf{x}_i, \mathbf{y}_{2,i}; \Theta, \eta_2) \right), \tag{1}$$

where $\mathbf{w} = [w_1, \ldots, w_n]^T$ is a pre-defined vector of each fixed weight $w_i$ assigned for each training sample $(\mathbf{x}_i, \mathbf{y}_{1,i}, \mathbf{y}_{2,i})$. We now consider training dynamic for GRW, where the weight $\mathbf{w}^{(t)}$ is the weight at $t$-th iteration:

$$\mathcal{L}_{\text{train}}(f) = \sum_i w_i^{(t)} \left( \mathcal{L}_{\text{cls}}(\mathbf{x}_i, \mathbf{y}_{1,i}; \Theta^{(t)}, \eta_1^{(t)}) + \mathcal{L}_{\text{reg}}(\mathbf{x}_i, \mathbf{y}_{2,i}; \Theta^{(t)}, \eta_2^{(t)}) \right), \tag{2}$$

In the regime of Neural Tangent Kernel (NTK) (Jacot et al., 2018), we have the following Proposition of an object detection network:

**Proposition 1.** *(NTRF approximation of DNNs.) When the width of neural networks goes infinite, the output of over-parameterized neural networks can be approximated as a linear function at iteration $t$:*

$$\hat{\mathbf{y}}_1^{(t)} = \psi^{(t)} \cdot \Theta^{(t)} \cdot \eta_1^{(t)}, \tag{3}$$

$$\hat{\mathbf{y}}_2^{(t)} = \psi^{(t)} \cdot \Theta^{(t)} \cdot \eta_2^{(t)}, \tag{4}$$

*where $\psi^{(t)} = \psi\left(\boldsymbol{X}, \Theta^{(t)}\right) \in \mathbb{R}^{n \times m}$ is the Neural Tangent Random Feature (NTRF) matrix (Cao & Gu, 2019) of $n$ training data $\boldsymbol{X}$, $\Theta \in \mathbb{R}^{m \times k}$ denotes the concatenation of all vectorized trainable parameters with size $m$, $\eta_1^{(t)} \in \mathbb{R}^k$ and $\eta_2^{(t)} \in \mathbb{R}^k$ project features into classification output $\hat{\mathbf{y}}_1^{(t)} \in \mathbb{R}^n$ and regression output $\hat{\mathbf{y}}_2^{(t)} \in \mathbb{R}^n$, respectively.*

We derive the following theorem and demonstrate that when $f$ is trained **simultaneously** by classification loss $\mathcal{L}_{\text{cls}}$ and regression loss $\mathcal{L}_{\text{reg}}$, vanilla GRW and ERM converged to the same solution. This demonstrates the failure of GRW in improving generalization abilities, which explains why previous algorithms do not achieve significant improvements.

**Theorem 1.** *(Failure of GRW on object detection.) If $\psi^{(0)} = \left[\psi_1^{(0)}, \ldots, \psi_n^{(0)}\right]^T$ is linearly independent, then when object detection network $\Theta, \eta_1, \eta_2$ simultaneously trained by 0-1 binary cross entropy loss and smooth L1 loss[1], as $t \to \infty$ and $\hat{\mathcal{L}}_{\text{train}} \to 0$, both GRW and ERM converge to the solution $\Theta^*, \eta_1^*, \eta_2^*$ that not depend on $\mathbf{w}$.*

---

[1]For simplicity, we consider binary classification here and one can easily derive the version for the multi-category classification following the same analysis. We also consider the interval $[-1, 1]$ for smooth L1 loss, which can be achieved by normalization on $\hat{\mathbf{y}}_2$.

Theorem 1 shows that if the initial NTRF matrix $\psi^{(0)}$, which depends on the training data $\boldsymbol{X}$ and the initial parameter $\Theta^{(0)}$, is linearly independent, then according to Cramer's Rule, there exists only one solution for minimizing empirical training risk. Proof of Theorem 1 can be found in Appendix A.1.

## 5 METHODOLOGY

### 5.1 FEATURE PERTURBATION

To cope with the issue of GRW, we propose certifiable feature perturbation (CFP), which aims to achieve single domain generalization through learning a robust network against perturbations and increasing the impact of $\mathbf{w}$ during training. Practically, we add Gaussian perturbation $\mathbf{m}^{(t)} \sim \mathcal{N}(0, \sigma^2 I), \mathbf{m}^{(t)} \in \mathbb{R}^k$ to the feature extracted by $\Theta^{(t)}$ and, in the NTK, the neural network with CFP is approximated as follows:

$$\hat{\mathbf{y}}_1^{(t)} = \left(\psi^{(t)} \cdot \Theta^{(t)} + \mathbf{1} \cdot \mathbf{m}^{(t)T}\right) \cdot \eta_1^{(t)}, \tag{5}$$

$$\hat{\mathbf{y}}_2^{(t)} = \left(\psi^{(t)} \cdot \Theta^{(t)} + \mathbf{1} \cdot \mathbf{m}^{(t)T}\right) \cdot \eta_2^{(t)}, \tag{6}$$

where $\mathbf{1} \cdot \mathbf{m}^{(t)T} = [\mathbf{m}^{(t)}, \ldots, \mathbf{m}^{(t)}]^T, \mathbf{1} \cdot \mathbf{m}^{(t)T} \in \mathbb{R}^{n \times k}$ is the sample-wise duplication of $\mathbf{m}^{(t)}$. With the new approximation, the presence of weight $\mathbf{w}$ during training can be guaranteed by the following theorem:

**Theorem 2.** *(Presence of weight* $\mathbf{w}$*.) When object detection network $\Theta, \eta_1, \eta_2$ with feature perturbations $\mathbf{m}$ trained by GRW, as $t \to \infty$, $\Theta^{(t)}, \eta_1^{(t)}, \eta_2^{(t)}$ is impacted by $\mathbf{w}^{(t)}$ and converge to $\Theta^*, \eta_1^*, \eta_2^*$ that depend on not only $\boldsymbol{X}, \Theta^{(0)}$ but also $\mathbf{w}^{(t)}$ for any t.*

Proof of Theorem 2 can be found in Appendix A.2. Theorem 2 demonstrates that the sample weights $\mathbf{w}$ should have an impact on the training dynamic, leading ERM and GRW to two different solutions. This overcomes the previous problem that GRW fails to provide a better solution than ERM.

### 5.2 CERTIFYING GENERALIZATION.

In this section, we demonstrate the effectiveness of CFP for improving single domain generalization ability in our algorithm framework. In a nutshell, we want to certify that if the loss of an object detection network with CFP is lower than the given classification loss $C_{\text{cls}}$ and the given regression loss $C_{\text{reg}}^2$, this remains true for certain perturbation sets $\mathcal{A}$. Firstly, following Ye et al. (2023), we define the expectation version of the $\mathcal{A}$-Generalizable for generalization:

**Definition 1.** *($\mathcal{A}$-Generalizable.) For a binary classification and regression problem, given a closed set $\mathcal{A}$, when $\frac{1}{n}\mathcal{L}_{\text{cls}}(\mathbf{x}, \mathbf{y}_1; \Theta, \eta_1) < C_{\text{cls}}$ and $\frac{1}{n}\mathcal{L}_{\text{reg}}(\mathbf{x}, \mathbf{y}_2; \Theta, \eta_2) < C_{\text{reg}}$, if for perturbation drawn from distribution $\mathbf{a} \in \mathcal{A}$, we have:*

$$\max_{\mathbf{a} \in \mathcal{A}} \mathbb{E}_{\mathbf{m} \sim \mathbf{a}} \left[\frac{1}{n}\mathcal{L}_{\text{cls}}(\mathbf{x}, \mathbf{y}_1, \mathbf{m}; \Theta, \eta_1)\right] < C_{\text{cls}}, \tag{7}$$

$$\max_{\mathbf{a} \in \mathcal{A}} \mathbb{E}_{\mathbf{m} \sim \mathbf{a}} \left[\frac{1}{n}\mathcal{L}_{\text{reg}}(\mathbf{x}, \mathbf{y}_2, \mathbf{m}; \Theta, \eta_2)\right] < C_{\text{reg}}, \tag{8}$$

*then the object detection network $f(\Theta, \eta_1, \eta_2)$ is $\mathcal{A}$-Generalizable.*

Based on Definition 1, the generalization ability of the proposed CARD can be guaranteed by the following theorem:

**Theorem 3.** *(Generalization of feature perturbation.) When an object detection network trained with Gaussian perturbation $\mathbf{m} \sim \mathcal{N}(0, \sigma^2 I)$, denoted as $f_{\mathbf{m}}(\Theta, \eta_1, \eta_2)$, if the classification loss $\frac{1}{n}\mathcal{L}_{\text{cls}}(\mathbf{x}, \mathbf{y}_1, \mathbf{m}; \Theta, \eta_1) < C_{\text{cls}}$ and the regression loss $\frac{1}{n}\mathcal{L}_{\text{reg}}(\mathbf{x}, \mathbf{y}_2, \mathbf{m}; \Theta, \eta_2) < C_{\text{reg}}$, then $f_{\mathbf{m}}(\Theta, \eta_1, \eta_2)$ is $\mathcal{A}$-Generalizable for $\mathcal{A} = \{\mathbf{a} : \delta \sim \mathbf{a}, \mu(\delta) = 0, \mathbb{V}[\delta] \leq \sigma^2 I\}$.*

$\mu(\cdot)$ is the mean function, and $\mathbb{V}[\cdot]$ is the variance function. Theorem 3 shows that when training with CARD, the converged model is $\mathcal{A}$-Generalizable. Proof can be found in Appendix A.3.

---

[2] Random predictions is like making predictions by flipping coins.

## 5.3 Algorithm Framework

Practically,estimating $\mathbf{w}$ in Equation (1) poses a challenge. Rather than directly applying reweighting during batch-wise training, we introduce a penalty term $\mathcal{L}_{\text{var}}$ based on the variance of batch-wise loss. This serves to steer the network towards "equally considering" each training sample, mitigating bias and, in effect, achieving implicit reweighting of the training samples:

$$\mathcal{L}_{\text{var}}(\mathbf{x}, \mathbf{y}_1, \mathbf{y}_2) = \frac{1}{n} \sum_i \left( \mathcal{L}_{\text{train}}(\mathbf{x}_i, \mathbf{y}_{1,i}, \mathbf{y}_{2,i}) - \frac{1}{n} \sum_k \mathcal{L}_{\text{train}}(\mathbf{x}_k, \mathbf{y}_{1,k}, \mathbf{y}_{2,k}) \right)^2 \tag{9}$$

This approach is supported by Krueger et al. (2021) and the variance penalty fits within the GRW. Both methods aim to eliminate the learning bias of DNNs by upweighting the samples that carry causal information ignored by the DNNs. On the other hand, Theorem 2 demonstrates that with the help of CFP, the modified algorithm should converge to different DNN parameters than ERM. Moreover, We demonstrate in Theorem 3 that CFP is robust against perturbations and generalizable under OoD scenarios, effectively harnessing the full potential of GRW to alleviate learning bias. These theoretical guarantees underpin a training algorithm for improved OoD generalization performances, distinguishing it from most previous efforts in S-DGOD, which rely solely on empirical results. The overall algorithm framework is shown in Algorithm 1 for training object detection networks for S-DGOD.

---

**Algorithm 1 CARD**: **C**ertifi**A**ble **R**eweighting for Single **D**omain Generalization Object **D**etection

---

**Require:** Training set $\mathcal{D}_{\text{train}}$, maximum of iterations $T$, learning rate $\beta$, penalty weight $\lambda_{\text{p}}$ and variance $\sigma^2$.
**Ensure:** $\mathcal{A}$-Generalizable object detection network $f$ with optimized parameters $\Theta^*, \eta_1^*, \eta_2^*$.
1: Initialize the detector network $f\left(\Theta^{(0)}, \eta_1^{(0)}, \eta_2^{(0)}\right)$ ;
2: **while** $t \leq T$ **do**
3:      **for** each $(\mathbf{x}, \mathbf{y}) \in \mathcal{D}_{\text{train}}$ **do**
4:          Extract the feature $\mathbf{z} = f\left(\mathbf{x}; \Theta^{(t)}\right) = \psi^{(0)} \Theta^{(t)}$ ;
5:          Generate the perturbation $\mathbf{m}^{(t)} \sim \mathcal{N}(0, \sigma^2 I)$ ;
6:          $\mathbf{z} \leftarrow \mathbf{z} + \mathbf{1} \cdot \mathbf{m}^{(t)T}$ ;
7:          Calculate empirical training loss $\hat{\mathcal{L}}_{\text{train}} = \mathcal{L}_{\text{cls}}\left(\mathbf{x}, \mathbf{y}; \Theta^{(t)}, \eta_1^{(t)}\right) + \mathcal{L}_{\text{reg}}\left(\mathbf{x}, \mathbf{y}; \Theta^{(t)}, \eta_2^{(t)}\right)$ ;
8:          Calculate variance penalty $\mathcal{L}_{\text{var}}\left(\mathbf{x}, \mathbf{y}; \Theta^{(t)}, \eta_1^{(t)}, \eta_2^{(t)}\right)$ according to Equation (9) ;
9:          $\Theta^{(t+1)} \leftarrow \Theta^{(t)} - \beta \cdot \nabla_{\Theta}(\hat{\mathcal{L}}_{\text{train}} + \lambda_{\text{p}} \mathcal{L}_{\text{var}})$ ;
10:         $\eta_1^{(t+1)} \leftarrow \eta_1^{(t)} - \beta \cdot \nabla_{\eta_1}(\hat{\mathcal{L}}_{\text{train}} + \lambda_{\text{p}} \mathcal{L}_{\text{var}})$ ;
11:         $\eta_2^{(t+1)} \leftarrow \eta_2^{(t)} - \beta \cdot \nabla_{\eta_2}(\hat{\mathcal{L}}_{\text{train}} + \lambda_{\text{p}} \mathcal{L}_{\text{var}})$ ;
12: Save the optimized network $f(\Theta^*, \eta_1^*, \eta_2^*)$ ;

---

## 6 Experiments

In this section, we first specify the experimental setting, then we conduct CARD and baseline methods on the challenging S-DGOD benchmarks for comparison. We also perform an ablation study to empirically substantiate our theoretical framework. Finally, we provide some visualizations to offer further insights into the superior performance of CARD. More experiments can be found in Appendix A.4, including effects of hyper-parameters in Appendix A.4.2 and robustness against adversarial attacks in Appendix A.4.3.

### 6.1 Experimental Setup

**Datasets.** For evaluation, we follow the experimental setting in Wu & Deng (2022). Our evaluation dataset comprises five urban-scene domains, each characterized by different time and weather conditions. These domains are as follows: Daytime-Sunny, Daytime-Foggy, Dusk-Rainy, Night-Sunny, and Night-Rainy. The images in the evaluation datasets are collected from multiple commonly-used benchmark datasets, including BDD100K (Yu et al., 2020), Cityscapes (Cordts et al., 2016), Foggy Cityscapes (Sakaridis et al., 2018), and Adverse-Weather datasets (Hassaballah et al., 2020). Specifically, the Daytime-Sunny domain contains a total of 27,708 images, with 19,395 allocated for train-

| Method | Daytime-Sunny | Daytime-Foggy | Dusk-Rainy | Night-Sunny | Night-Rainy | Average |
|---|---|---|---|---|---|---|
| Faster R-CNN (Ren et al., 2015) | 54.2 | 38.6 | 30.6 | 38.7 | 14.2 | 35.3 |
| IBN-Net (Pan et al., 2018) | 53.9 | **38.9** | 32.2 | 38.2 | 15.4 | 35.7 |
| SW (Pan et al., 2019) | 54.0 | 35.4 | 27.6 | 35.9 | 12.6 | 33.1 |
| IterNorm (Huang et al., 2019) | 53.9 | 38.1 | 32.3 | 38.4 | 15.2 | 35.6 |
| ISW (Choi et al., 2021) | 52.5 | 35.9 | 34.6 | 33.0 | 16.2 | 34.4 |
| ASRNorm (Fan et al., 2021) | 54.1 | 36.6 | 30.4 | 37.8 | 14.4 | 34.7 |
| CDSD (Wu & Deng, 2022) | 55.2 | 33.9 | 30.2 | 39.2 | 14.0 | 34.5 |
| CARD (Ours) | **59.5** | 36.7 | **35.5** | **44.6** | **16.8** | **38.6** |

Table 1: **Single domain generalization object detection results.** All algorithms are initialized by the ImageNet pre-trained weights and are trained on the Daytime-Sunny domain then tested on the other four domains. Average results are calculated using the five domain results. All algorithms are implemented on the mmdetection code suite (Chen et al., 2019). The numbers in bold or underlined denote the highest and the second performance, respectively. The results demonstrate that our approach is robust against domain shifts and achieves the SOTA average S-DGOD performance.

| Method | Init | Quantity | D-S | D-F | D-R | N-S | N-R | Average |
|---|---|---|---|---|---|---|---|---|
| CLIPGap (Vidit et al., 2023) | CLIP | 400M | 58.4 | **44.3** | 34.1 | 41.3 | 16.8 | 39.0 |
| Faster R-CNN (Ren et al., 2015) | ImageNet | 1.2M | 54.2 | 38.6 | 30.6 | 38.7 | 14.2 | 35.3 |
| Faster R-CNN (Ren et al., 2015) | ImageNet + MS COCO | 1.5M | 57.7 | 37.1 | 34.2 | 44.2 | 16.6 | 38.0 |
| CARD (Ours) | ImageNet | 1.2M | 59.5 | 36.7 | 35.5 | 44.6 | 16.8 | 38.6 |
| CARD (Ours) | ImageNet + MS COCO | 1.5M | **59.9** | 37.3 | **35.9** | **45.3** | **17.4** | **39.2** |

Table 2: **Comparisons with different initialization strategies.** "Init" column is the dataset used in pre-training and "Quantity" is the number of images in datasets. D, F, R, N, and S represent Daytime, Foggy, Rainy, Night, and Sunny, respectively. These results demonstrate that pre-trained datasets can significantly impact the generalization performance, while our proposed CARD is less reliant on pre-trained knowledge to achieve supreme OoD generalization performance.

ing and 8,313 for test. The remaining four domains are exclusively reserved for test. The Daytime-Foggy domain comprises 3,775 images, the Dusk-Rainy comprises 3,501 images, the Night-Sunny comprises 26,158 images and the Night-Rainy domain comprises 2,494 images. To be consistent with Wu & Deng (2022), we target seven important object categories, including bus, bike, car, motorbike, person, rider, and truck. Samples of this dataset are shown in Figure 2.

**Baselines.** We follow the S-DGOD benchmarks proposed by Wu & Deng (2022), which contains five well-developed algorithms for improving generalization ability, including IBN-Net (Pan et al., 2018), Switchable Whitening (SW) introduced by Pan et al. (2019), Iterative Normalization (IterNorm) introduced by Huang et al. (2019), RobustNet (ISW) introduced by Choi et al. (2021), Adversarially Adaptive Normalization (ASRNorm) introduced by Fan et al. (2021), and Cyclic-Disentangled Self-Distillation (CDSD) introduced by Wu & Deng (2022). We also present the performance of Faster R-CNN (Ren et al., 2015) as all algorithms are implemented based on Faster R-CNN. Additionally, we further compare our method with the current SOTA method, CLIPGap (Vidit et al., 2023). However, the authors highlight that the improvement is primarily attributed to the pre-trained weights obtained from CLIP (Radford et al., 2021), which utilizes an extremely large-scale dataset. In contrast, other baseline algorithms developed on the S-DGOD benchmarks Wu & Deng (2022) only have access to much smaller datasets (Deng et al., 2009). To ensure fairness, we further compare with CLIPGap under different pre-training strategies, as shown in Table 2.

**Evaluation metric.** In all experiments, we use the Mean Average Precision (mAP) to evaluate methods' performances and report the AP of each class. Specifically, we follow the PASCAL VOC evaluation metric (Everingham et al., 2010) and report the mAP with a 0.5 intersection over union (IoU) threshold, where a prediction is considered as true positive if its IoU score with the ground truth label is more than 0.5.

**Implemetation details.** All baseline algorithms use the popular two-stage Faster R-CNN detector Ren et al. (2015) with ResNet-101 backbone (He et al., 2016) and Feature Pyramid Network

| Method | Dusk-Rainy | | | | | | | | Night-Sunny | | | | | | | |
|---|---|---|---|---|---|---|---|---|---|---|---|---|---|---|---|---|
| | Bus | Bike | Car | Motor | Person | Rider | Truck | mAP | Bus | Bike | Car | Motor | Person | Rider | Truck | mAP |
| FR | 36.3 | 22.3 | 62.7 | 14.9 | 27.3 | 14.8 | 35.8 | 30.6 | 37.5 | 36.0 | 61.7 | 20.9 | 45.5 | 27.8 | 41.3 | 38.7 |
| IBN-Net | 38.1 | 27.4 | 63.3 | 7.8 | 30.9 | 19.8 | 38.2 | 32.2 | 36.2 | 34.7 | 60.9 | 18.4 | 45.2 | 30.8 | 41.1 | 38.2 |
| SW | 33.7 | 20.0 | 58.5 | 7.8 | 22.8 | 17.5 | 32.8 | 27.6 | 34.8 | 32.1 | 57.6 | 18.2 | 43.9 | 26.0 | 38.4 | 35.9 |
| IterNorm | 39.3 | 23.3 | 63.1 | 10.4 | 30.6 | 19.8 | 39.7 | 32.3 | 38.1 | 30.2 | 61.0 | 20.1 | 46.9 | 31.0 | 41.6 | 38.4 |
| ISW | 39.9 | 22.7 | 65.0 | 15.5 | 38.3 | 21.6 | 39.0 | 34.6 | 32.8 | 26.8 | 59.4 | 13.4 | 40.6 | 23.9 | 34.1 | 33.0 |
| ASRNorm | 36.7 | 23.1 | 62.8 | 6.4 | 27.9 | 16.8 | 39.3 | 30.4 | 37.6 | 35.9 | 60.1 | 11.2 | 46.1 | 29.8 | 43.6 | 37.8 |
| CDSD | 37.9 | 20.5 | 59.1 | 13.8 | 24.6 | 16.3 | 39.3 | 30.2 | 36.7 | 35.7 | 62.0 | 23.2 | 41.5 | 33.5 | 41.5 | 39.2 |
| CARD (Ours) | 43.2 | 26.1 | 67.8 | 12.2 | 32.4 | 21.4 | 45.5 | 35.5 | 44.0 | 40.2 | 67.4 | 25.7 | 52.2 | 35.3 | 47.8 | 44.6 |

Table 3: **Per-class results on Dusk-Rainy and Night-Sunny.** FR denotes Faster R-CNN. The numbers in bold or underlined denote the highest and the second performance, respectively.

| Method | CFP | Daytime-Sunny | Daytime-Foggy | Dusk-Rainy | Night-Sunny | Night-Rainy | Average |
|---|---|---|---|---|---|---|---|
| ERM | ✗ | 54.2 | **38.6** | 30.6 | 38.7 | 14.2 | 35.3 |
| ERM | ✓ | 58.1 | 35.9 | 33.9 | 41.7 | 16.3 | 37.2 |
| GRW | ✗ | 55.4 | 36.0 | 31.4 | 40.1 | 14.9 | 35.6 |
| GRW | ✓ | **59.5** | 36.7 | **35.5** | **44.6** | **16.8** | **38.6** |

Table 4: **Ablation study.** These results show that the vanilla GRW marginally outperforms ERM and the effectiveness of CFP in improving OoD performance. Moreover, GRW works well with CFP and significantly achieves the optimal average performance.

(FPN) introduced by Lin et al. (2017a) [3]. At the beginning of training, if not specified, networks are initialized by the ImageNet pre-trained weights. We train all models on the Daytime-Sunny source domain for 24 epochs for full convergence. We set the $\lambda_g$ to 1.0 and the $\sigma^2$ to 0.0001. We apply the Stochastic Gradient Descent optimizer with the 0.02 learning rate. All experiments are conducted on a server with 8 GPUs with 4 samples per GPU.

## 6.2 COMPARISON WITH THE STATE OF THE ART

We first report the overall results on the S-DGOD benchmark and then per-class results.

**Overall S-DGOD results.** Table 1 shows the mAP results on all domains, including Daytime-Sunny, Daytime-Foggy, Dusk-Rainy, Night-Sunny, and Night-Rainy, in which Daytime-Sunny is for training. As shown in Table 1, our proposed CARD significantly improves the average performance to 38.6% compared with the baselines algorithm, i.e., IBN-Net, which achieves 35.7%. This suggests CARD learns the generalizable features to make predictions. Furthermore, we compare CARD with the current SOTA algorithm, CLIPGap, under the same experimental protocol introduced by Vidit et al. (2023) in the five domains. Results are shown in Table 2 and our proposed CARD surpasses CLIPGap by 0.2% with significantly less pre-trained knowledge. These results demonstrate the supreme ability of CARD in learning generalizable features and achieving SOTA OoD performance.

**Per-class results.** Table 3 lists the per-class results of CARD and baselines in the Dusk-Rainy, and Night-Sunny, which are two extremely challenging target domains. For more per-class results, please refer to Appendix A.4.1. The results show that CARD outperforms baselines in detecting buses, cars, and trucks with the highest accuracy while showing comparable accuracy in other classes. These demonstrate the effectiveness of CARD and also indicate that CARD is effective in the popular life-critical application, autonomous driving (AD), where the DNN-based AD system significantly degenerates in new environments, suffering a lot from the domain shift. Moreover, CARD achieves SOTA performance in all classes in the Night-Sunny domain, where it is especially challenging to clearly identify completed objects even for humans. This demonstrates the generalization ability of CARD as well as the robustness against low-light conditions.

## 6.3 ABLATION STUDY

We conduct an ablation study to understand the effect of GRW and CFP separately. As shown in Table 4, GRW without (w/o) CFP marginally surpasses ERM w/o CFP by 0.3% in terms of average

---

[3]The results later updated by Wu & Deng (2022) in the released codes use the FPN. To align with Wu & Deng (2022), all used baselines are re-implemented. Baseline results are improved with the FPN.

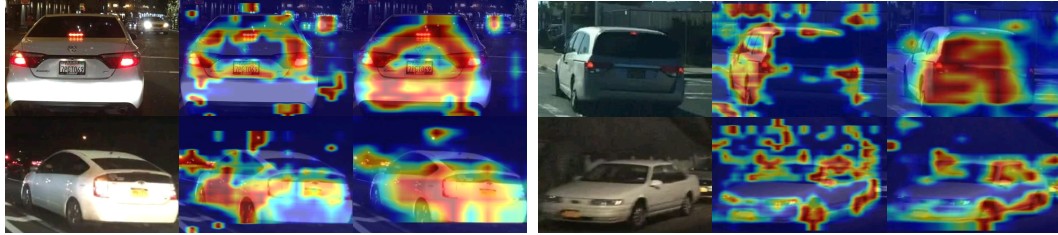

Figure 3: **EigenCam Visualization.** *Left*: Ground truth. *Middle*: Baselines. *Right*: CARD. These figures show that CARD is able to make predictions based on the object-related pixels, while baseline focuses less on the object itself.

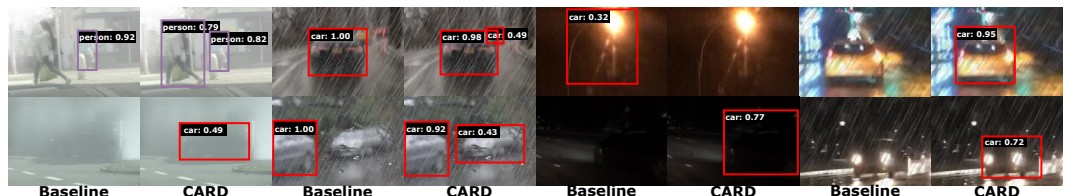

Figure 4: **Inference results (category: confidence) on the four target domains.** Every two columns, where the left column is the results of baseline and the right is CARD's, from left to right represent the results on Daytime-Foggy, Dusk-Rainy, Night-Sunny, and Night-Rainy. Compared with baselines, CARD is able to detect the lost objects and accurately recognize objects and contexts. Better view in zoom-in mode.

performance. This empirically demonstrates that the vanilla GRW is hard to outperform ERM, consistent with the proposition in Theorem 1. On the contrary, the inclusion of CFP ("w. CFP") in the GRW framework elevates the average performance to 38.6%, surpassing GRW w/o CFP by 3.0%. Additionally, ERM w. CFP also surpasses ERM w/o CFP by 1.9%. These results underscore the generalizability of CFP against challenging domain shifts, aligning with the theoretical insights presented in Theorem 3. Moreover, CFP enhances the performance of both ERM and GRW, with the improvement on GRW surpassing that on ERM. This suggests that CFP synergizes effectively with GRW in OoD scenarios, harnessing the full potential of GRW to alleviate learning bias.

### 6.4 VISUALIZATION

**EigenCam.** Figure 3 presents the EigenCam (Muhammad & Yeasin, 2020) results of baselines and CARD. As the figure shows, our proposed CARD makes predictions based on the object-related regions, while baselines rely on the background-related features. These show that CARD effectively mitigates the impact of spurious correlations and makes predictions grounded on causal information.

**Inference results.** Figure 4 presents some inference comparisons of baselines and CARD on the four target domains and demonstrates the supreme generalization ability of CARD under extremely challenging scenarios. More inference results can be found in Appendix A.5.

## 7 CONCLUSION

In this paper, we introduce a novel approach named Certifiable Reweighting (CARD), which is OoD-aware with theoretical guarantees, for single domain generalization object detection (S-DGOD). First of all, we consider the effective technique for improving OoD performance, Generalizable Reweighting (GRW), which reweights the training samples during training. We identify and demonstrate that the vanilla GRW struggles to surpass the performance of Empirical Risk Minimization (ERM). To address this challenge, we introduce Certifiable Feature Perturbation (CFP), which is designed to train a robust neural network against random perturbations and helps GRW learn a generalizable network. Our extensive experiments show our proposed CARD achieves SOTA performance compared with previous SOTA baselines in the challenging S-DGOD benchmarks.

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

# A  APPENDIX

## A.1  PROOF OF THEOREM 1

We first restate the linear approximation of DNNs:

**Proposition 1.** *(NTRF approximation of DNNs.) When the width of neural networks goes infinite, the output of over-parameterized neural networks can be approximated as a linear function at iteration $t$:*

$$\hat{\mathbf{y}}_1^{(t)} = \psi^{(t)} \cdot \Theta^{(t)} \cdot \eta_1^{(t)}, \tag{3}$$

$$\hat{\mathbf{y}}_2^{(t)} = \psi^{(t)} \cdot \Theta^{(t)} \cdot \eta_2^{(t)}, \tag{4}$$

*where $\psi^{(t)} = \psi\left(\boldsymbol{X}, \Theta^{(t)}\right) \in \mathbb{R}^{n \times m}$ is the Neural Tangent Random Feature (NTRF) matrix (Cao & Gu, 2019) of $n$ training data $\boldsymbol{X}$, $\Theta \in \mathbb{R}^{m \times k}$ denotes the concatenation of all vectorized trainable parameters with size $m$, $\eta_1^{(t)} \in \mathbb{R}^k$ and $\eta_2^{(t)} \in \mathbb{R}^k$ project features into classification output $\hat{\mathbf{y}}_1^{(t)} \in \mathbb{R}^n$ and regression output $\hat{\mathbf{y}}_2^{(t)} \in \mathbb{R}^n$, respectively.*

Then we restate the theorem as follows:

**Theorem 1.** *(Failure of GRW on object detection.) If $\psi^{(0)} = \left[\psi_1^{(0)}, \ldots, \psi_n^{(0)}\right]^T$ is linearly independent, then when object detection network $\Theta, \eta_1, \eta_2$ simultaneously trained by 0-1 binary cross entropy loss and smooth L1 loss[4], as $t \to \infty$ and $\hat{\mathcal{L}}_{\text{train}} \to 0$, both GRW and ERM converge to the solution $\Theta^*, \eta_1^*, \eta_2^*$ that not depend on $\mathbf{w}$.*

*Proof.* According to Lee et al. (2019), NTRF matrix $\psi^{(t)}$ changes very little during training, and following Pezeshki et al. (2021), we set $\psi^{(t)} \equiv \psi^{(0)}$. As $\psi^{(0)} = \left[\psi_1^{(0)}, \ldots, \psi_n^{(0)}\right]^T$ is linearly independent, according to Cramer's Rule, there is exactly one $\Theta^*, \eta_1^*, \eta_2^*$ so that:

$$\mathbf{y}_1 = \psi^{(t)}\Theta^*\eta_1^* = \psi^{(0)}\Theta^*\eta_1^*, \tag{10}$$

$$\mathbf{y}_2 = \psi^{(t)}\Theta^*\eta_2^* = \psi^{(0)}\Theta^*\eta_2^*. \tag{11}$$

Recall the empirical training loss:

$$\hat{\mathcal{L}}_{\text{train}}(\mathbf{x}, \mathbf{y}; \Theta, \eta_1, \eta_2) = \mathcal{L}_{\text{cls}}(\mathbf{x}, \mathbf{y}; \Theta, \eta_1) + \mathcal{L}_{\text{reg}}(\mathbf{x}, \mathbf{y}; \Theta, \eta_2). \tag{12}$$

For regression, we have:

$$\left|\psi^{(0)}\left(\Theta^{(t)}\eta_2^{(t)} - \Theta^*\eta_2^*\right)\right| = \left|\left(\psi^{(0)}\Theta^{(t)}\eta_2^{(t)} - \mathbf{y}_2\right) - \left(\psi^{(0)}\Theta^*\eta_2^* - \mathbf{y}_2\right)\right|$$
$$\leq \left|\psi^{(0)}\Theta^{(t)}\eta_2^{(t)} - \mathbf{y}_2\right| + \left|\psi^{(0)}\Theta^*\eta_2^* - \mathbf{y}_2\right|. \tag{13}$$

Together with

$$\mathcal{L}_{\text{reg}}(\mathbf{x}, \mathbf{y}_2; \Theta, \eta_2) = \frac{1}{2}\left(\psi^{(0)}\Theta^{(t)}\eta_2^{(t)} - \mathbf{y}_2\right)^T \left(\psi^{(0)}\Theta^{(t)}\eta_2^{(t)} - \mathbf{y}_2\right), \tag{14}$$

as $t \to \infty$ and $\hat{\mathcal{L}}_{\text{train}} \to 0$, we have $\mathcal{L}_{\text{reg}} \to 0$. Therefore, we have:

$$\left|\psi^{(0)}\Theta^{(t)}\eta_2^{(t)} - \mathbf{y}_2\right| + \left|\psi^{(0)}\Theta^*\eta_2^* - \mathbf{y}_2\right| \to \mathbf{0}, \tag{15}$$

$$\left|\psi^{(0)}\left(\Theta^{(t)}\eta_2^{(t)} - \Theta^*\eta_2^*\right)\right| \to \mathbf{0}, \tag{16}$$

where $\mathbf{0}$ is the all-zeros vector with size $n$. This indicates that $\Theta^{(t)}$ and $\eta_2^{(t)}$ converge to $\Theta^*$ and $\eta_2^*$, respectively, because $\eta_2 \in \mathbb{R}$. Similarly, for classification at time $t$, we have:

$$\mathcal{L}_{\text{cls}}(\mathbf{x}, \mathbf{y}_1; \Theta^{(t)}, \eta_1^{(t)}) = -\mathbf{1}^T \cdot \left(\mathbf{y}_1 \log\left(\psi^{(0)}\Theta^{(t)}\eta_1^{(t)}\right) + (1 - \mathbf{y}_1)\log\left(1 - \psi^{(0)}\Theta^{(t)}\eta_1^{(t)}\right)\right), \tag{17}$$

---

[4]For simplicity, we consider binary classification here and one can easily derive the version for the multi-category classification following the same analysis. We also consider the interval $[-1, 1]$ for smooth L1 loss, which can be achieved by normalization on $\hat{\mathbf{y}}_2$.

where $\mathbf{1}$ is the all-ones vector with size $n$. As $t \to \infty$ and $\hat{\mathcal{L}}_{\text{train}} \to 0$, we have $\mathcal{L}_{\text{cls}} \to 0$, thus, $\Theta^{(t)}$ and $\eta_1^{(t)}$ converge to $\Theta^*$ and $\tilde{\eta}_1$, respectively. We have:

$$\mathbf{y}_1 = \psi^{(0)} \Theta^* \tilde{\eta}_1. \tag{18}$$

Together with

$$\mathbf{y}_1 = \psi^{(0)} \Theta^* \eta_1^*, \tag{19}$$

then $\tilde{\eta}_1 = \eta_1^*$, which means $\eta_1^{(t)}$ converges to $\eta_1^*$. From the above derivation, we can see that $\Theta, \eta_1, \eta_2$ trained by ERM converge to $\Theta^*, \eta_1^*, \eta_2^*$. We consider adding sample weights $\mathbf{w} = [w_1, \ldots, w_n]^T$ on Equation (12), we have:

$$\mathcal{L}_{\text{train}}(\mathbf{x}, \mathbf{y}; \Theta, \eta_1, \eta_2) = \mathbf{w}^T \left( \mathcal{L}_{\text{cls}}(\mathbf{x}, \mathbf{y}; \Theta, \eta_1) + \mathcal{L}_{\text{reg}}(\mathbf{x}, \mathbf{y}; \Theta, \eta_2) \right). \tag{20}$$

Following the same derivation above, we can conclude that $\Theta, \eta_1, \eta_2$ trained by GRW also converge to $\Theta^*, \eta_1^*, \eta_2^*$. This demonstrates that as long as $t \to \infty$ and $\hat{\mathcal{L}}_{\text{train}} \to 0$, when $\psi^{(0)} = \left[ \psi_1^{(0)}, \ldots, \psi_n^{(0)} \right]^T$ is linearly independent, both GRW and ERM converge to the same solution $\Theta^*, \eta_1^*, \eta_2^*$. $\qquad \square$

## A.2  PROOF OF THEOREM 2

We first restate the theorem:

**Theorem 2.** *(Presence of weight $\mathbf{w}$.) When object detection network $\Theta, \eta_1, \eta_2$ with feature perturbations $\mathbf{m}$ trained by GRW, as $t \to \infty$, $\Theta^{(t)}, \eta_1^{(t)}, \eta_2^{(t)}$ is impacted by $\mathbf{w}^{(t)}$ and converge to $\Theta^*, \eta_1^*, \eta_2^*$ that depend on not only $\mathbf{X}, \Theta^{(0)}$ but also $\mathbf{w}^{(t)}$ for any $t$.*

The proof is shown as follows:

*Proof.* When training with GRW, $\Theta$ is updated as the following way:

$$\Theta^{(t+1)} = \Theta^{(t)} - \beta \sum_i w_i^{(t)} \left( \nabla_\Theta \mathcal{L}_{\text{cls}}(\mathbf{x}_i, \mathbf{y}_{1,i}; \Theta^{(t)}, \eta_1^{(t)}) + \nabla_\Theta \mathcal{L}_{\text{reg}}(\mathbf{x}_i, \mathbf{y}_{2,i}; \Theta^{(t)}, \eta_2^{(t)}) \right) \tag{21}$$

$$= \Theta^{(t)} - \beta \sum_i w_i^{(t)} \psi^{(t)} \left( \frac{\mathbf{y}_{1,i} - \hat{\mathbf{y}}_{1,i}}{\hat{\mathbf{y}}_{1,i}(1 - \hat{\mathbf{y}}_{1,i})} \eta_1^{(t)} + (\hat{\mathbf{y}}_{2,i} - \mathbf{y}_{2,i}) \eta_2^{(t)} \right) \tag{22}$$

$$= \Theta^{(t)} - \beta \sum_i w_i^{(t)} \psi^{(0)} \left( \frac{\mathbf{y}_{1,i} - \hat{\mathbf{y}}_{1,i}}{\hat{\mathbf{y}}_{1,i}(1 - \hat{\mathbf{y}}_{1,i})} \eta_1^{(t)} + (\hat{\mathbf{y}}_{2,i} - \mathbf{y}_{2,i}) \eta_2^{(t)} \right) \tag{23}$$

$$= \Theta^{(t)} - \beta \sum_i w_i^{(t)} \psi^{(0)} c_i^t \tag{24}$$

$$c_i^t = \frac{\mathbf{y}_{1,i} - \hat{\mathbf{y}}_{1,i}}{\hat{\mathbf{y}}_{1,i}(1 - \hat{\mathbf{y}}_{1,i})} \eta_1^{(t)} + (\hat{\mathbf{y}}_{2,i} - \mathbf{y}_{2,i}) \eta_2^{(t)}, c_i^t \in \mathbb{R}^k. \tag{25}$$

Thus, we can derive the formulation for $\Theta^{(t+1)}$:

$$\Theta^{(t+1)} - \Theta^{(0)} = -\beta \sum_j \sum_i w_i^{(j)} \psi^{(0)} c_i^j. \tag{26}$$

This shows that $w_i^{(t)}$ should impact the learning dynamic of $\Theta$. As $t \to \infty$, $\Theta^{(t+1)}$ converges to $\Theta^*$, and we can have:

$$\Theta^* = \Theta^{(0)} - \beta \sum_j \sum_i w_i^{(j)} \psi^{(0)} c_i^j. \tag{27}$$

This shows that $\Theta^*$ depends on $w_i^{(t)}$, $\boldsymbol{X}$ and $\Theta^{(0)}$. Similarly, the update of $\eta_1$ and $\eta_2$ can be formalized as:

$$\eta_1^{(t+1)} = \eta_1^{(t)} - \beta \sum_i w_i^{(t)} \left( \nabla_{\eta_1} \mathcal{L}_{\text{cls}}(\mathbf{x}_i, \mathbf{y}_{1,i}; \Theta^{(t)}, \eta_1^{(t)}) + \nabla_{\eta_1} \mathcal{L}_{\text{reg}}(\mathbf{x}_i, \mathbf{y}_{2,i}; \Theta^{(t)}, \eta_2^{(t)}) \right) \quad (28)$$

$$= \eta_1^{(t)} - \beta \sum_i w_i^{(t)} \frac{\mathbf{y}_1 - \hat{\mathbf{y}}_1}{\hat{\mathbf{y}}(1 - \hat{\mathbf{y}})} \left( \psi^{(0)} \Theta^{(t)} + \mathbf{m}_i^{(t)} \right), \quad (29)$$

$$\eta_2^{(t+1)} = \eta_2^{(t)} - \beta \sum_i w_i^{(t)} \left( \nabla_{\eta_2} \mathcal{L}_{\text{cls}}(\mathbf{x}_i, \mathbf{y}_{1,i}; \Theta^{(t)}, \eta_1^{(t)}) + \nabla_{\eta_2} \mathcal{L}_{\text{reg}}(\mathbf{x}_i, \mathbf{y}_{2,i}; \Theta^{(t)}, \eta_2^{(t)}) \right) \quad (30)$$

$$= \eta_2^{(t)} - \beta \sum_i w_i^{(t)} (\hat{\mathbf{y}}_2 - \mathbf{y}_2) \left( \psi^{(0)} \Theta^{(t)} + \mathbf{m}_i^{(t)} \right). \quad (31)$$

as $t \to \infty$, $\eta_1^{(t+1)}$ and $\eta_2^{(t+1)}$ converges to $\eta_1^*$ and $\eta_2*$:

$$\eta_1^* = \eta_1^{(0)} - \beta \sum_j \sum_i w_i^{(j)} \frac{\mathbf{y}_1 - \hat{\mathbf{y}}_1}{\hat{\mathbf{y}}(1 - \hat{\mathbf{y}})} \left( \psi^{(0)} \Theta^{(j)} + \mathbf{m}_i^{(j)} \right), \quad (32)$$

$$\eta_2^* = \eta_2^{(0)} - \beta \sum_j \sum_i w_i^{(j)} (\hat{\mathbf{y}}_2 - \mathbf{y}_2) \left( \psi^{(0)} \Theta^{(j)} + \mathbf{m}_i^{(j)} \right). \quad (33)$$

These show that $\eta_1^*, \eta_2^*$ should also depend on $w_i^{(t)}$ for any $t$ and demonstrate the presence of $w_i^{(t)}$ in the training process. $\qquad \square$

### A.3 PROOF OF THEOREM 3

We first recall the definition of $\mathcal{A}$-Generalizable:

**Definition 1.** *($\mathcal{A}$-Generalizable.) For a binary classification and regression problem, given a closed set $\mathcal{A}$, when $\frac{1}{n}\mathcal{L}_{\text{cls}}(\mathbf{x}, \mathbf{y}_1; \Theta, \eta_1) < C_{\text{cls}}$ and $\frac{1}{n}\mathcal{L}_{\text{reg}}(\mathbf{x}, \mathbf{y}_2; \Theta, \eta_2) < C_{\text{reg}}$, if for perturbation drawn from distribution $\mathbf{a} \in \mathcal{A}$, we have:*

$$\max_{\mathbf{a} \in \mathcal{A}} \mathbb{E}_{\mathbf{m} \sim \mathbf{a}} \left[ \frac{1}{n} \mathcal{L}_{\text{cls}}(\mathbf{x}, \mathbf{y}_1, \mathbf{m}; \Theta, \eta_1) \right] < C_{\text{cls}}, \quad (7)$$

$$\max_{\mathbf{a} \in \mathcal{A}} \mathbb{E}_{\mathbf{m} \sim \mathbf{a}} \left[ \frac{1}{n} \mathcal{L}_{\text{reg}}(\mathbf{x}, \mathbf{y}_2, \mathbf{m}; \Theta, \eta_2) \right] < C_{\text{reg}}, \quad (8)$$

*then the object detection network $f(\Theta, \eta_1, \eta_2)$ is $\mathcal{A}$-Generalizable.*

We then restate the theorem:

**Theorem 3.** *(Generalization of feature perturbation.) When an object detection network trained with Gaussian perturbation $\mathbf{m} \sim \mathcal{N}(0, \sigma^2 I)$, denoted as $f_{\mathbf{m}}(\Theta, \eta_1, \eta_2)$, if the classification loss $\frac{1}{n}\mathcal{L}_{\text{cls}}(\mathbf{x}, \mathbf{y}_1, \mathbf{m}; \Theta, \eta_1) < C_{\text{cls}}$ and the regression loss $\frac{1}{n}\mathcal{L}_{\text{reg}}(\mathbf{x}, \mathbf{y}_2, \mathbf{m}; \Theta, \eta_2) < C_{\text{reg}}$, then $f_{\mathbf{m}}(\Theta, \eta_1, \eta_2)$ is $\mathcal{A}$-Generalizable for $\mathcal{A} = \{\mathbf{a} : \delta \sim \mathbf{a}, \mu(\delta) = 0, \mathbb{V}[\delta] \leq \sigma^2 I\}$.*

The proof is shown as follows:

*Proof.* Consider the classification loss using binary cross entropy function:

$$\frac{1}{n} \mathbb{E}_{\mathbf{m} \sim \mathcal{N}(0, \sigma^2 I)} \left[ \mathcal{L}_{\text{cls}}(\mathbf{x}, \mathbf{y}_1, \mathbf{m}; \Theta^*, \eta_1^*) \right] < C_{\text{cls}}. \quad (34)$$

For clarity, we assume the category label $\mathbf{y}_1 = \mathbf{1}$:

$$\frac{1}{n} \mathbb{E}_{\mathbf{m} \sim \mathcal{N}} \left[ \mathcal{L}_{\text{cls}}(\mathbf{x}, \mathbf{y}_1, \mathbf{m}; \Theta^*, \eta_1^*) \right] < C_{\text{cls}} \quad (35)$$

$$\Longleftrightarrow \frac{1}{n} \mathbb{E}_{\mathbf{m} \sim \mathcal{N}} \left[ \mathbf{1}^T \cdot f_{\mathbf{m}}(\mathbf{x}, \mathbf{y}_1; \Theta^*, \eta_1^*) \right] > \frac{1}{2} \quad (36)$$

$$\Longleftrightarrow \frac{1}{n} \mathbb{E}_{\mathbf{m} \sim \mathcal{N}(0, \sigma^2 I)} \left[ \mathbf{1}^T \cdot (\psi^{(0)} \Theta^* + \mathbf{1} \cdot \mathbf{m}^{(t)T}) \eta_1^* \right] > \frac{1}{2} \quad (37)$$

$$\Longleftrightarrow \frac{1}{n} \mathbf{1}^T \psi^{(0)} \Theta^* \eta_1^* > \frac{1}{2}. \quad (38)$$

For some perturbation $\delta \in \mathbb{R}^k$ drawn from some distribution $\mathbf{a}$ with zero mean, we have:

$$\frac{1}{n}\mathbb{E}_{\delta \sim \mathbf{a}}\left[\mathbf{1}^T \cdot (\psi^{(0)}\Theta^* + \mathbf{1} \cdot \delta^T)\eta_1^*\right] = \frac{1}{n}\mathbf{1}^T\psi^{(0)}\Theta^*\eta_1^* > \frac{1}{2}. \tag{39}$$

Consider the regression loss using the smooth L1 function:

$$\frac{1}{n}\mathbb{E}_{\mathbf{m}\sim\mathcal{N}(0,\sigma^2 I)}\left[\mathcal{L}_{\text{reg}}\left(\mathbf{x}, \mathbf{y}_2, \mathbf{m}; \Theta^*, \eta_2^*\right)\right] \tag{40}$$

$$=\frac{1}{n}\mathbb{E}_{\mathbf{m}\sim\mathcal{N}(0,\sigma^2 I)}\left[\frac{1}{2}\left(f_{\mathbf{m}}(\mathbf{x}, \mathbf{y}_2; \Theta^*, \eta_2^*) - \mathbf{y}_2\right)^T \left(f_{\mathbf{m}}(\mathbf{x}, \mathbf{y}_2; \Theta^*, \eta_2^*) - \mathbf{y}_2\right)\right] \tag{41}$$

$$=\frac{1}{2n}\mathbb{E}_{\mathbf{m}\sim\mathcal{N}(0,\sigma^2 I)}\left[\left(\left(\psi^{(0)}\Theta^* + \mathbf{1} \cdot \mathbf{m}^T\right)\eta_2^* - \mathbf{y}_2\right)^T \left(\left(\psi^{(0)}\Theta^* + \mathbf{1} \cdot \mathbf{m}^T\right)\eta_2^* - \mathbf{y}_2\right)\right] \tag{42}$$

$$=\frac{1}{2n}\mathbb{E}_{\mathbf{m}\sim\mathcal{N}(0,\sigma^2 I)}\left[\left(\psi^{(0)}\Theta^*\eta_2^* - \mathbf{y}_2 + \mathbf{1} \cdot \mathbf{m}^T\eta_2^*\right)^T \left(\psi^{(0)}\Theta^*\eta_2^* - \mathbf{y}_2 + \mathbf{1} \cdot \mathbf{m}^T\eta_2^*\right)\right] \tag{43}$$

$$=\frac{1}{2n}\left(\psi^{(0)}\Theta^*\eta_2^* - \mathbf{y}_2\right)^T \left(\psi^{(0)}\Theta^*\eta_2^* - \mathbf{y}_2\right) + \frac{1}{2n}\mathbb{E}_{\mathbf{m}\sim\mathcal{N}(0,\sigma^2 I)}\left[\eta_2^{*T}\mathbf{m}\mathbf{m}^T\eta_2^*\right] \tag{44}$$

$$=\frac{1}{2n}\left(\psi^{(0)}\Theta^*\eta_2^* - \mathbf{y}_2\right)^T \left(\psi^{(0)}\Theta^*\eta_2^* - \mathbf{y}_2\right) + \frac{1}{2n}\mathbb{E}_{\mathbf{m}\sim\mathcal{N}(0,\sigma^2 I)}\left[\eta_2^{*T}\mathbf{m}\right]^2 \tag{45}$$

$$=\frac{1}{2n}\left(\psi^{(0)}\Theta^*\eta_2^* - \mathbf{y}_2\right)^T \left(\psi^{(0)}\Theta^*\eta_2^* - \mathbf{y}_2\right) + \frac{1}{2n}\eta_2^{*T}\mathbb{V}_{\mathbf{m}\sim\mathcal{N}(0,\sigma^2 I)}\left[\mathbf{m}\right]\eta_2^* \tag{46}$$

$$=\frac{1}{2n}\left(\psi^{(0)}\Theta^*\eta_2^* - \mathbf{y}_2\right)^T \left(\psi^{(0)}\Theta^*\eta_2^* - \mathbf{y}_2\right) + \frac{\sigma^2}{2n}\eta_2^{*T}\eta_2^* < C_{\text{reg}}. \tag{47}$$

$$\tag{48}$$

For the perturbation $\delta \sim \mathbf{a}$, if elements in $\delta$ are pairwise independent, we have

$$\frac{1}{n}\mathbb{E}_{\delta \sim \mathbf{a}}\left[\mathcal{L}_{\text{reg}}\left(\mathbf{x}, \mathbf{y}_2, \delta; \Theta^*, \eta_2^*\right)\right] < C_{\text{reg}}, \tag{49}$$

$$\Longleftrightarrow \frac{1}{2n}\left(\psi^{(0)}\Theta^*\eta_2^* - \mathbf{y}_2\right)^T \left(\psi^{(0)}\Theta^*\eta_2^* - \mathbf{y}_2\right) + \frac{1}{2n}\eta_2^{*T}\mathbb{V}_{\delta \sim \mathbf{a}}\left[\delta\right]\eta_2^* < C_{\text{reg}}, \tag{50}$$

when $\mathbb{V}[\delta] \leq \sigma^2 I$, where $\mathbb{V}[\cdot]$ is the variance function. $\square$

## A.4 MORE EXPERIMENT RESULTS

### A.4.1 PER-CLASS RESULTS

Table 5 and Table 6 list the per-class results on Daytime-Foggy and Night-Rainy, respectively. CARD shows comparable car detection performance on Daytime-Foggy and achieves SOTA performance on Night-Rainy. This indicates our proposed method can be further implemented in the challenging autonomous driving application, which suffers a lot from the domain shift.

| Method | Bus | Bike | Car | Motor | Person | Rider | Truck | mAP |
|---|---|---|---|---|---|---|---|---|
| Faster R-CNN (Ren et al., 2015) | **35.7** | 35.5 | 58.2 | 33.9 | **40.0** | 42.7 | 24.1 | 38.6 |
| IBN-Net (Pan et al., 2018) | 34.3 | 35.4 | 57.5 | **34.3** | 39.8 | **43.8** | **27.3** | **38.9** |
| SW (Pan et al., 2019) | 31.1 | 32.0 | 52.8 | 31.9 | 34.8 | 41.3 | 23.8 | 35.4 |
| IterNorm (Huang et al., 2019) | 32.8 | **35.7** | 57.3 | 32.9 | 39.3 | 43.4 | 25.1 | 38.1 |
| ISW (Choi et al., 2021) | 31.2 | 33.6 | 52.8 | 31.7 | 39.8 | 40.7 | 21.6 | 35.9 |
| ASRNorm (Fan et al., 2021) | 30.5 | 34.7 | 52.3 | 31.2 | 38.9 | 43.4 | 25.0 | 36.6 |
| CDSD (Wu & Deng, 2022) | 30.0 | 29.7 | 52.4 | 30.3 | 33.4 | 40.1 | 21.5 | 33.9 |
| CARD (Ours) | 33.0 | 31.1 | 58.0 | 31.6 | 38.6 | 40.0 | 24.9 | 36.7 |

Table 5: **Per-class results on Daytime-Foggy.**

### A.4.2 HYPER-PARAMETERS

Table 7 lists the performance on the S-DGOD benchmarks with different hyper-parameters and the results show that CARD achieves the optimal OoD performance when $\lambda_p$ and $\sigma^2$ are set to 1.0 and 0.0001.

| Method | Bus | Bike | Car | Motor | Person | Rider | Truck | mAP |
|---|---|---|---|---|---|---|---|---|
| Faster R-CNN (Ren et al., 2015) | 21.2 | 9.6 | 33.9 | 0.2 | 11.8 | 1.5 | 21.0 | 14.2 |
| IBN-Net (Pan et al., 2018) | 23.1 | 10.1 | 34.4 | 0.2 | 12.4 | 4.8 | **22.6** | 15.4 |
| SW (Pan et al., 2019) | 16.3 | 9.1 | 26.2 | 0.4 | 10.6 | 9.1 | 16.8 | 12.6 |
| IterNorm (Huang et al., 2019) | 24.5 | 10.7 | 32.8 | 0.6 | 13.9 | 6.3 | 17.5 | 15.2 |
| ISW (Choi et al., 2021) | 22.8 | 10.2 | 33.5 | **1.3** | **15.7** | **11.4** | 18.6 | 16.2 |
| ASRNorm (Fan et al., 2021) | 21.9 | **11.6** | 30.3 | 0.1 | 11.7 | 4.5 | 20.5 | 14.4 |
| CDSD (Wu & Deng, 2022) | 19.4 | 9.5 | 31.4 | 0.2 | 11.1 | 9.7 | 16.6 | 14.0 |
| CARD (Ours) | **27.0** | 9.4 | **37.6** | 0.4 | 12.9 | 9.1 | 21.1 | **16.8** |

Table 6: **Per-class results on Night-Rainy.**

| Method | $\lambda_p$ | $\sigma^2$ | Daytime-Sunny | Daytime-Foggy | Dusk-Rainy | Night-Sunny | Night-Rainy | Average |
|---|---|---|---|---|---|---|---|---|
| CARD (Ours) | 2.0 | 0.0001 | 58.9 | 36.2 | 34.9 | 44.2 | 16.5 | 38.1 |
| CARD (Ours) | 1.0 | 0.0001 | **59.5** | **36.7** | **35.5** | **44.6** | **16.8** | **38.6** |
| CARD (Ours) | 0.5 | 0.0001 | 58.7 | 35.9 | 35.2 | 44.0 | 16.7 | 38.1 |
| CARD (Ours) | 1.0 | 0.01 | 57.3 | 34.7 | 34.1 | 42.9 | 16.0 | 37.0 |
| CARD (Ours) | 1.0 | 0.001 | 58.5 | 35.1 | 35.3 | 43.6 | **16.9** | 37.9 |
| CARD (Ours) | 1.0 | 0.0001 | **59.5** | **36.7** | **35.5** | **44.6** | 16.8 | **38.6** |

Table 7: **Results of different hyper-parameters.**

### A.4.3 ADVERSARIAL ATTACK

We conduct the adversarial attack experiments on Table 8 using FGSM (Goodfellow et al., 2014). As the results show, our CARD surpasses ERM by 4.5% under the attack by FGSM. This demonstrates that CARD is more robust against adversarial attacks than ERM baselines.

| Method | Daytime-Sunny | Daytime-Foggy | Dusk-Rainy | Night-Sunny | Night-Rainy | Average |
|---|---|---|---|---|---|---|
| ERM | 54.2 | **38.6** | 30.6 | 38.7 | 14.2 | 35.3 |
| ERM+FGSM | 50.1 | 33.9 | 28.3 | 35.6 | 13.8 | 32.3 |
| CARD (Ours) | **59.5** | 36.7 | **35.5** | **44.6** | **16.8** | **38.6** |
| CARD (Ours)+FGSM | 56.1 | 34.9 | 34.2 | 42.1 | 16.6 | 36.8 |

Table 8: **Results under adversarial attack.**

### A.5 INFERENCE RESULTS

Figure 5 presents more inference results of CARD, which show the supreme performance compared with baselines.

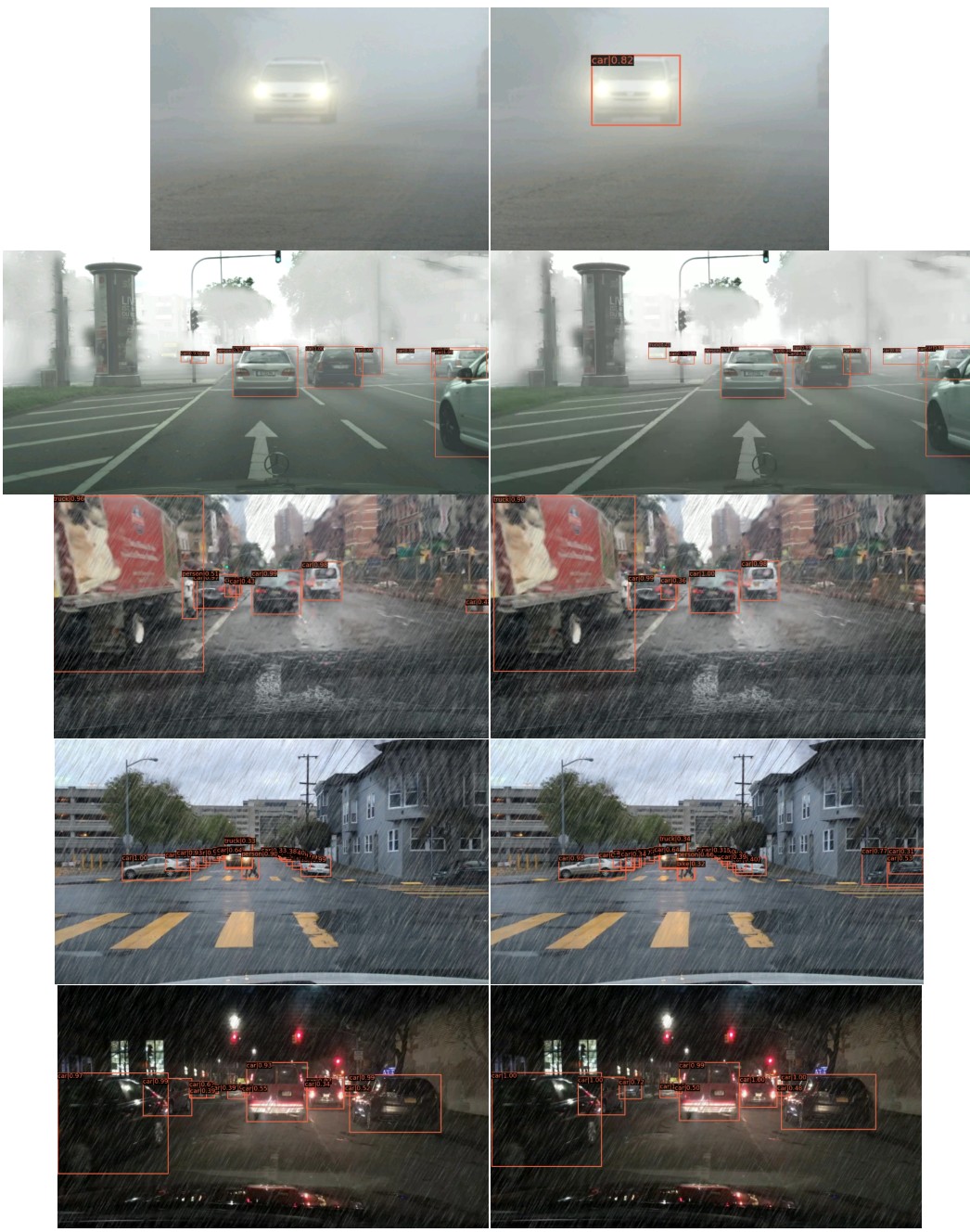

Figure 5: **More inference results.** The left column is the predictions of baselines and the right column is the predictions of CARD.

