# OpenReview forum: "CARD: Certifiable Reweighting for Single Domain Generalization Object Detection"
_ICLR.cc/2024/Conference — ICLR 2024 Conference Withdrawn Submission_

### Official Review · Reviewer_suAi · 2023-10-14

**Soundness:** 2 fair
**Presentation:** 2 fair
**Contribution:** 2 fair
**Rating:** 5
**Confidence:** 5

**Summary:**

This paper focuses on the task of single-domain generalized object detection, which aims to generalize the detector trained based on one single source domain to multiple unseen domains with different styles. This task is challenging yet practical. To this end, this paper explores a classic and widely-used approach, Generalizable Reweighting (GRW), which iteratively reweightes the training samples to improve generalization performance. This paper gives sufficient theoretical analysis. And experimental results show the effectiveness of the proposed method.

**Strengths:**

The theoretical analysis of this paper is sufficient. Experimental results demonstrate the effectiveness of the proposed method.

**Weaknesses:**

1. The motivation of this paper is somewhat unclear. In the first paragraph of Introduction Section, the authors introduce the task of single-domain generalized object detection. However, the second paragraph directly indicates that generalizable reweighting is a classical yet effective technique. To the best of my knowledge, there exist some works for S-DGOD. The authors should sufficiently introduce these works and analyze their corresponding problems, which is beneficial for further understanding the motivation of this paper.

2. To the best of my knowledge, there exists a type of technique that aims to utilize active learning to improve the generalization ability of models. I am not clear about the difference between active learning and generalizable reweighting. The authors should give more interpretations.

3. In Eq. (5) and (6), directly adding Gaussian perturbation to the extracted feature may destroy the semantic information, as it is hard to control the perturbation degree. Besides, the authors introduce a penalty term based on the variance of batch-wise loss. However, computing batch-wise loss may introduce additional computational costs. The authors should give more interpretations and analysis.

4. In Fig. 3, the shown images only contain one object, i.e., car, which is somewhat simple. The authors should show more different examples containing multiple objects. Besides, the main contribution of this paper is to propose a new loss function. However, this paper does not show the optimization details. The authors should show training curves. Finally, it is better to show failure detection results and give corresponding interpretations, which is helpful for understanding the proposed method.

**Questions:**

The authors should interpret the difference between active learning and the proposed method.

---

### Official Review · Reviewer_7drN · 2023-11-06

**Soundness:** 2 fair
**Presentation:** 2 fair
**Contribution:** 2 fair
**Rating:** 3
**Confidence:** 5

**Summary:**

This paper analyzes the problem of generalizable reweighting (GRW) by comparing to ERM and finds that GRW obtains the same solution as ERM even under different paths. Then, this paper presents a certifiable feature perturbation (CFP) which makes GRW have different soultion from ERM and even surpasses it against additional perturbations. The authors name this proposal as CARD and works for single domain generalizable object detection (SDGOD). Experiments show the proposed CARD achieves SOTA.

**Strengths:**

+ It is interesting to prove the failure of GRW in domain generalization by theoretical support and achieves the same solution as ERM, which does not depend on the sample weight.
+ It is nice to introduce the feature perturbation into the GRW and makes GRW and ERM  unequally. The writting is easy to follow and clearly describes the basic idea and framework.

**Weaknesses:**

+ Introducing feature perturbation makes sure the solution of GRW depends on the sample weight, but does not guarantee the superiority of CFP+GRW theoretically. Although this may not be important as experiments can show that, there may be other better ways than CFP to achieve that. This is not discussed. More ways can be provided with comparisons.
+ Although the authors claim that for multi-category classification, this does hold in footnote, I also suggest analyze that rather than only considering binary classification. Object detection in this submission is commonly a multi-class problem.
+ This submission claims the SOTA performance. However, several more references on single domain generalizable object detectors are not mentioned [1,2,3], and even the number of references in the subtopic is small. Therefore, the contribution of this work is weak.
+ More domain generalization methods in image classification can be compared for S-DGOD.
+ There are multiple variables not defined in the equations. In algorithm 1, how is z used?  which is not clarifed. The writting can be further improved.
+ From the whole algorithm, the feature perturabation is only used, and the training is just GRW. The technical novelty is limited.

[1] Domain-invariant disentangled network for generalizable object detection, ICCV 2021.
[2] Multi-view Adversarial Discriminator: Mine the Non-causal Factors for Object Detection in Unseen Domains. CVPR 2023.
[3] Randomized Spectrum Transformations for Adapting Object Detector in Unseen Domains, TIP 2023.

**Questions:**

1. In implementation, you just take the perturbated features as the input of GRW based detector?
2. The physical intuition of Eq.9 can be further explained.
3. In Fig.1 right, the flat minimum is demontrated. However, an actual picture during optimization can be given to show that.

---

### Official Review · Reviewer_Y71C · 2023-11-23

**Soundness:** 2 fair
**Presentation:** 2 fair
**Contribution:** 2 fair
**Rating:** 5
**Confidence:** 2

**Summary:**

The paper shows theoretically that Generalizable Reweighting (GRW) converges to the same solution as Empirical Risk Minimization (ERM) in the context of Single Domain Generalizable Object Detection (SDGOD). The paper then proposes the CARD algorithm to improve GRW with Certifiable Feature Perturbation (CFP). Experiments shows it brings improvements over vanilla GRW.

**Strengths:**

- The paper contributes theoretically by showing the convergence of GRW and ERM to the same solution, independent of sample weight w.
- The proposed CARD algorithm via CFP, effectively improves vanilla GRW in the realm of SDGOD.

**Weaknesses:**

- The technical contribution seems somewhat limited. CFP is adding Gaussian perturbation to the feature map during training, which is a commonly used strategy to improve roburstness of the trained model.
- The numbers for other methods in the paper appears different from the cited paper. In CLIPGap (Vidit et al., 2023) Table 1, the corresponding baseline numbers are very differernt from the numbers in this paper's Table 1. CLIPGap's number aligns with the original numbers from CDSD (Wu & Deng, 2022). This discrepancy raises questions about methodology and fairness of comparison. The authors should provide clarification on whether the baselines were re-implemented and why such comparison is fair.
- The discussion of the theory's applicability to various loss functions is limited. Further exploration, such as its compatibility with other classification/regression losses or other applications with both regression and classification losses, would enhance the paper's contribution.
- The author shows that CFP makes GRW to converges to a different solution than ERM. But different solution doesn't necessarily imply better solution. The author should explain why this would improve model performance.

**Questions:**

- The term "vanilla GRW" needs further clarification. Which specific algorithm is used? How is w determined during GRW training?
- The role of Neural Tangent Kernel (NTK) in the derivations is not thoroughly explained. Could the authors provide more insight into its significance and impact on the overall findings?

---

### Official Review · Reviewer_W6J9 · 2023-11-25

**Soundness:** 2 fair
**Presentation:** 2 fair
**Contribution:** 2 fair
**Rating:** 3
**Confidence:** 3

**Summary:**

This work studied the single-domain generalization object detection problem, where data is available from only one domain and the rest of the domains (target domains) are being utilized to evaluate the generalization ability. The authors have built some theoretical results to establish the generalization guarantee for a robust object detection model. Specifically, the authors aim to improve generalizable reweighting with certifiable feature perturbation and call it CARD which gains state-of-the-art performance in urban scene benchmarks.

**Strengths:**

+ Theoretical analysis along with the experimental findings are satisfactory for this study.
+ This paper addresses an important problem, single-domain generalization object detection is a challenging problem.
+ Paper is easy to follow and well written.

**Weaknesses:**

+ It is unclear what factors are core to the object detection problem, rather this can be applied to the image classification task as well.
+ Perturbation as Gaussian is not a new thing to add to improve generalizability. Did the authors try adding other perturbations as mentioned in [1] for comparison? This will give the influence of specific perturbations in the relevant task.
+ How this task is impacted by adding feature level perturbation and input level perturbation in comparison, and how does it affect the overall training process?
+ Equation 9 is not well explained and needs more clarity to show its significance.

[1] Benchmarking Neural Network Robustness to Common Corruptions and Perturbations, ICLR 2019

**Questions:**

Q: Providing a comparison with transformer-based architectures is beneficial. It is widely believed and studied that transformers are more generalizable in nature.